# HER2-Positive Gastric Cancer and Antibody Treatment: State of the Art and Future Developments

**DOI:** 10.3390/cancers16071336

**Published:** 2024-03-29

**Authors:** Magdalena K. Scheck, Ralf D. Hofheinz, Sylvie Lorenzen

**Affiliations:** 1Klinik und Poliklinik für Innere Medizin III, Klinikum rechts der Isar der TU München, 81675 Munich, Germany; magdalena.scheck@mri.tum.de; 2Mannheim Cancer Center, Universitätsklinikum Mannheim, 68167 Mannheim, Germany; ralf-dieter.hofheinz@medma.uni-heidelberg.de

**Keywords:** HER2, gastric cancer, trastuzumab, T-DXd, targeted therapy

## Abstract

**Simple Summary:**

Biomarker-guided therapies offer the possibility for targeted cancer therapies. For gastric cancer, HER2 plays a pivotal role. To date, with trastuzumab and trastuzumab-deruxtecan, two HER2-targeting options exist for the treatment of metastatic HER2-positive gastric cancer. However, these therapies rarely lead to long-lasting results as resistance mechanisms and heterogeneity limit their success. This review gives an overview of the current standard-of-care treatment of HER2-positive gastric cancer and new developments in the field.

**Abstract:**

Despite a decreasing incidence in Western countries, gastric cancer is among the most common cancer subtypes globally and is associated with one of the highest tumor-related mortality rates. Biomarkers play an increasing role in the treatment against gastric cancer. HER2 was one of the first biomarkers that found its way into clinical practice. Since the ToGA trial, trastuzumab has been part of first-line palliative chemotherapy in metastatic or unresectable gastric cancer. HER2-targeting agents, such as the tyrosine kinase inhibitor lapatinib, the antibody drug conjugate (ADC) trastuzumab-emtansine or dual HER2 inhibition (pertuzumab and trastuzumab), have been investigated in the second-line setting but led to negative study results. More recently, the ADC trastuzumab-deruxtecan was authorized after the failure of trastuzumab-based treatment. However, further improvements in HER2-directed therapy are required as resistance mechanisms and HER2 heterogeneity limit the existing treatment options. This review aims to give an overview of the current standard-of-care HER2-directed therapy in gastric cancer, as well as its challenges and future developments.

## 1. Introduction

Gastric cancer still imposes a major health threat, with over one million new cases in 2020 and 769,000 deaths, ranking it the fourth leading cause of cancer-related deaths globally [1]. Human epidermal growth factor receptor-2 (HER2), also referred to as HER2/neu or ERBB2, is present in about 20% of gastric cancer cases [2,3].

HER2 is encoded on chromosome 17q21 and is part of the EGFR family, which also includes EGFR/HER1, HER3 and HER4 [4]. The ligand of the tyrosine kinase receptor is still not identified. HER2 forms either homodimers or heterodimers with other members of the EGFR family receptors [5]. HER2 homodimers can lead to ligand-independent activation in case of HER2 overexpression [6]. Heterodimers, however, rely on receptor-specific ligand binding [7]. In particular, the HER2/HER3 heterodimer was found to produce a potent mitogenic signal induced by its ligand, heregulin [8].

HER2 activation leads to downstream signaling via the PI3K/AKT/mTOR pathway or the RAS/MEK/MAPK pathways, inducing cell cycle progression, cell growth, proliferation of the cell, metastasis and inhibiting cell death [4,7,9].

Amplification or overexpression of HER2 can be found in various cancer entities. In gastric cancer, ERBB2 alterations are mainly present in the chromosomal unstable (CIN) subtype [10]. These aberrations mainly consist of missense mutations (74%) and, only to a lesser extent, insertions (22%) or fusions (0.7%) [11]. HER2 expression is more common in intestinal subtype according to Lauren’s classification, as well as proximal gastric cancer or gastroesophageal junction cancer, metastatic disease (especially liver metastasis) and lymph node invasion. Its prognostic relevance is still up for debate [4,5,7,12].

This review aims to give an overview of the current standard-of-care treatment as well as new therapeutic approaches to targeting HER2-positive gastric cancer.

## 2. Treatment of Metastatic HER2-Positive Gastric Cancer

Compared to the broad therapeutic options that are available for breast cancer, HER2-targeting therapies for gastric cancer remain limited [13].

### 2.1. Introduction of Trastuzumab

The monoclonal antibody trastuzumab inhibits downstream signaling by directly binding the juxtamembrane region (extracellular domain IV) of the HER2 receptor. This leads to endocytosis of the receptor, a blockade of either the constitutive active forms or the dimerization of HER2, and antibody-dependent cell-mediated cytotoxicity (ADCC) [5,7]. The ToGA study, including patients with gastric or gastroesophageal junction adenocarcinoma, was the first study to implement HER2-directed therapy with trastuzumab into the treatment of metastatic HER2-positive gastric cancer [14]. Trastuzumab plus cisplatin and capecitabin or fluorouracil improved overall survival (OS 13.8 vs. 11.1 months, HR 0.74), progression-free survival (PFS 6.7 vs. 5.5 months, HR 0.71) and objective response rate (ORR 47% vs. 35%) compared to chemotherapy alone in 584 treated patients. An exploratory analysis of the ToGA trial revealed that prolongation of OS correlated with HER2 expression. In HER2 highly expressing tumors (IHC 2+/ISH-positive or IHC3+), an OS of 16 months compared to 11.8 months for chemotherapy alone was achieved (HR 0.65). Importantly, HER2 expression but not amplification levels correlated with a better response to trastuzumab, and HER2 testing is therefore firstly based on immunohistochemistry [2]. Patients received six cycles (about 3.5 months) of trastuzumab plus chemotherapy before trastuzumab maintenance was initiated. Various smaller studies and case reports have described long-term responses under trastuzumab maintenance [15,16]. Recently, a retrospective biomarker analysis revealed that patients with a prolonged PFS under trastuzumab maintenance have higher PD-L1 CPS scores, which correlate with increased CD4+ memory T cells. The tumor mutational burden, ERBB2 score or resistance-associated genetic alterations (see below) did not differ between short-term and long-term responders [17]. Attempts to further improve maintenance strategies have not led to success so far. A higher maintenance dose (8 mg/kg vs. 6 mg/kg) in the HELOISE trial did not increase efficacy (HR 1.24) [18]. The addition of fluoropyrimidines to trastuzumab maintenance showed no prolongation of OS (15.2 vs. 17.0 months for trastuzumab alone) or PFS (5.1 months in both cases) in the large retrospective AGEO trial [19]. Of note, the reintroduction of initial chemotherapy upon progression achieved a longer OS than standard second-line regimens in the AGEO trial and might be a valuable option to preserve later treatment lines.

The results of the ToGA trial led to the authorization of trastuzumab in the first-line treatment of metastatic or unresectable HER2-positive gastric cancer combined with capecitabine or 5-fluorouracil (5-FU) and cisplatin. In the ToGA trial, 88% of the patients received capecitabine and cisplatin [14]. The current ESMO guidelines recommend a combination of trastuzumab and platinum/fluoropyrimidine-based chemotherapy, whereas oxaliplatin and cisplatin, as well as 5-FU, capecitabin and S-1, are stated as possible combination partners [20]. Real-world data of the German observational study HerMES revealed that only half of 364 included patients were treated in-label [21]. Other common chemotherapy regimens were FLOT, monotherapy with 5-FU/capecitabine and FOLFOX. PFS and OS ranged from 6.4 to 9.5 months and 12.3 to 15.8 months, respectively. In a meta-analysis, Ter Veer et al. showed that the doublet chemotherapy backbone of oxaliplatin and 5-FU/capecitabine prolonged OS in comparison to the ToGA regimen (cisplatin and 5-FU, HR 0.75) and reduced toxicity [22]. Cisplatin plus S-1 caused less hand–foot syndrome, but showed no survival benefit compared to ToGA. Furthermore, a triplet cytotoxic backbone could not improve survival but increased toxicity, while monotherapy with cisplatin or capecitabine provided insufficient results compared to the ToGA regimen. The authors claimed that a doublet therapy containing oxaliplatin is favorable over the ToGA regimen with cisplatin [22].

### 2.2. Trastuzumab and Checkpoint Inhibition

Recently, the placebo-controlled phase III KEYNOTE-811 study evaluated the addition of immune checkpoint inhibition (ICI) to standard first-line therapy with trastuzumab and chemotherapy for HER2-positive esophagogastric cancer [23]. The study demonstrated a significant and clinically meaningful improvement regarding tumor response rate (ORR) for ICI plus trastuzumab/chemotherapy [23]. Based on these results, the FDA granted approval for pembrolizumab in first-line chemotherapy with trastuzumab.

Survival data were presented at the ESMO 2023 [24]. Although there was an advantage in PFS (10.0 vs. 8.1 months; HR 0.73) and OS (20.0 vs. 16.9 months; HR 0.84) for all tumors, the subgroup analysis revealed that only PD-L1-positive tumors benefit from additional ICI (PFS 10.9 vs. 7.3 months; HR 0.71). The proportion of PD-L1-positive tumors in this study was high (CPS ≥ 1 in 88%), which might be due to co-expression of HER2 and PD-L1 [3]. In comparison, among HER2-negative tumors in the CHECKMATE-649 study, about 82% had a positive PD-L1 score (PD-L1 CPS expression ≥ 1) [25]. The FDA therefore restricted approval to PD-L1-positive tumors (CPS ≥ 1). In August 2023, pembrolizumab was also authorized by the EMA for tumors with CPS ≥1. Pembrolizumab plus trastuzumab and fluoropyrimidine/platinum-based chemotherapy is therefore a new standard-of-care regimen for PD-L1- and HER2-positive unresectable or metastatic gastric and gastroesophageal junction adenocarcinoma.

### 2.3. Beyond Progression to Trastuzumab

Upon progression to trastuzumab, the combination of trastuzumab and paclitaxel was proven ineffective in the Japanese T-ACT trial, with similar PFS (3.7 vs. 3.2 months) and OS (10 months in both arms) compared to paclitaxel alone (see Table 1) [26]. However, the combination of trastuzumab and standard second-line therapy consisting of the VEGF inhibitor ramucirumab and paclitaxel showed promising anti-tumor activity even after progression to trastuzumab in first-line therapy. An ORR of 54% and a DCR of 96% were reported in the single-arm Her-RAM study [27].

The antibody–drug conjugate (ADC) **trastuzumab-deruxtecan** (Enhertu, T-DXd) is the first new therapeutic agent targeting HER2 to enter standard therapy for metastatic HER2-positive gastric cancer since the approval of trastuzumab. Antibody–drug conjugates (ADCs) link a cytotoxic drug to a monoclonal antibody by a chemical bond and therefore provide cytotoxicity as well as target specificity, stability and favorable pharmacokinetics [28]. T-DXd combines trastuzumab with a novel topoisomerase I inhibitor (exatecan mesylate) that has more potent efficacy than conventional SN-38, the active metabolite of irinotecan. With an antibody-to-drug ratio of 8:1, T-DXd has a higher payload than most ADCs. The linker consists of a tetrapeptide which is decomposed by cytosolic lysosomal enzymes expressed in tumor cells, releasing its cytotoxic load into the tumor cell. The binding activity to HER2 of T-DXd is comparable to unconjugated HER2-targeting agents. Besides its ADCC activity, T-DXd inhibits Akt phosphorylation and induces DNA damage [29]. Due to the membrane permeability of exatecan mesylate, T-DXd exhibits a bystander killing effect on surrounding tumor cells, which helps to overcome HER2 heterogeneity (see below) [30,31]. For metastatic breast cancer, T-DXd is already an available option after failure of trastuzumab in HER2-positive cancer and, since 2022, also in HER2-low expressing cancers (see below) [32].

T-DXd in gastric cancer was first tested in the Asian phase II DESTINY-Gastric01 trial. Upon the failure of at least two lines of therapy (including previous therapy with trastuzumab), T-DXd achieved better objective response rates (ORR 51% vs. 14%; *p* < 0.001) and confirmed objective response rates (i.e., an ORR lasting ≥ 4 weeks, cORR 43% vs. 12%) as well as longer OS (12.5 vs. 8.4 months) and PFS (5.6 vs. 3.5 months) than conventional chemotherapy (irinotecan or paclitaxel). Based on these data, T-DXd was approved by the FDA for HER2-positive metastatic gastric cancer [33]. Subgroup analysis revealed that HER2 3+ tumors benefit more from T-DXd than HER2 2+/FISH+ tumors, which is in line with in vitro results indicating that the anti-tumor efficacy of T-DXd is dependent on HER2 expression levels [31,33].

The EMA authorized T-DXd based on the results of the DESTINY-Gastric02 trial, which confirmed the efficacy of T-DXd in Western patients after the failure of trastuzumab-containing first-line therapy [34]. T-DXd showed a confirmed objective response of 42%, a median OS of 12.1 months and a median PFS of 5.6 months, which is consistent with the data from the DESTINY-Gastric01 study. Even though the current approval is granted without requiring re-biopsies to confirm HER2 status, histological reassessment should be discussed due to possible downregulation of HER2 after trastuzumab-containing therapy (see below).

Overall, the side effects of T-DXd were manageable, but severe treatment-emergent adverse events (CTCAE ≥ 3) occurred in over half of patients. The most common TEAEs were nausea and myelosuppression, which was addressed by dose reduction. The current NCCN guidelines classify T-DXd as a highly emetogenic substance and recommend triple prophylaxis consisting of NK1 receptor antagonist, 5-HT3 receptor antagonist and dexamethasone, which should be combined with olanzapine if necessary [35]. Interstitial lung disease (ILD) and pneumonitis are notable risks during T-DXd therapy, of which 10% were reported in each of the DESTINY-Gastric01 and 02 trials, with two cases ending fatally in DESTINY-Gastric02 [33,34]. Of note, the onset of ILD was late, within a median of 84.5 (DESTINY-Gastric01) and 80.5 (DESTINY-Gastric02) days after the start of therapy. Appropriate education on possible side effects and a close monitoring thereof are crucial during T-DXd therapy.

New therapeutic approaches and combinations with T-DXd are currently being assessed. DESTINY-Gastric04 (NCT04704934) compares T-DXd to the current second-line standard of ramucirumab and paclitaxel in a phase III trial. T-DXd plus chemotherapy and/or ICI is also under investigation as first-line therapy for metastatic HER2-positive gastric cancer in the phase Ib/II DESTINY-Gastric03 study (NCT04379596). Furthermore, T-DXd expresses activity in metastatic HER2-low breast cancer, and current ESMO expert consensus statements recommend the use of T-DXd in metastatic HER2-low breast cancer (IHC1+ or IHC2+/ISH−) after failure of prior CDK4/6 inhibitor therapy and at least one previous line of chemotherapy [36,37]. For gastric cancer, a small study with 21 patients recently provided preliminary evidence of the efficacy of T-DXd in IHC1+ (confirmed ORR 9.5%) or IHC2+/ISH− (confirmed ORR 26.3%) gastric cancer [38]. The ongoing DESTINY-Gastric03 trial will also partly include HER2-low gastric cancer.

Altogether, trastuzumab and T-DXd are, to date, the only HER2-targeting treatment options available for gastric cancer. In the metastatic setting, these two drugs are part of standard-of-care regimens (see Figure 1).

## 3. Resistance to Trastuzumab

Even though the introduction of trastuzumab and, more recently, T-DXd can be seen as a great success in the treatment of HER2-positive gastric cancer, it has to be considered that long-term responses are only achieved in a minority of patients. Resistance to trastuzumab presents as primary resistance or occurs in a relatively short amount of time as secondary resistance (time to progression in ToGA trial 7.1 months) [7,14]. To further overcome such resistance, it is crucial to understand the underlying mechanisms. These can be divided into (a) masking, downregulation or cleavage of the receptor, (b) interaction or cross-signaling with other receptors and (c) aberrations in downstream pathways.

Masking of HER2 was shown by membrane-associated glycoprotein MUC4 and hyaluronan, which prevents the binding of trastuzumab. The use of 4-MU, a hyaluronan synthase inhibitor, improved the binding of trastuzumab. Furthermore, the hyaluronan receptor CD44 plays a role in HER2 downregulation. The siRNA-mediated suppression of CD44 results in decreased internalization of trastuzumab [39]. Downregulation of the HER2 receptor upon progression to trastuzumab is described in 30 to 60% of the patients and occurs preferably in IHC 2+ scored tumors [40,41,42]. Janjigian et al. recently showed in next-generation sequencing data that the level of ERBB2 amplification is linked to the response to trastuzumab. Upon failure of trastuzumab, ERBB2 amplification was lost in resistant tumors [43]. Moreover, cleavage of the HER2 receptor results in a 95 kDa membrane-associated fragment (p95HER2) with increased activity [5].

Various interactions of HER2 with other receptors are described, serving as escape mechanisms for HER2-targeting agents. Members of the EGFR family such as EGFR or HER3 are upregulated or form heterodimers with HER2 [5,44,45]. In breast cancer cells, interactions and cross-signaling with other receptors like MET and heterodimers with insulin-like growth factor (IGFR1) were found. Importantly, in gastric cancer cells, IGFR1 can induce PI3K signaling upon its upregulation in a HER2-independent manner [5,46]. The upregulation of other tyrosine kinases such as the proto-oncogene Src or the receptor tyrosine kinase FGFR3 also contribute to the resistance against HER2-directed drugs [45,47,48]. A small proportion of HER2-positive tumors also show EGFR and/or cMET overexpression. Concomitant overexpression of receptor tyrosine kinases (RTKs) is associated with a poor prognosis, with the worst results being observed in triple (HER2+/MET+/EGFR+)-positive gastric cancer [49].

A significant alteration and/or constitutive activation of multiple pathways can be detected in trastuzumab-resistant cell lines. In a proteomic analysis, Liu et al. found that the mTOR, Wnt, p53, metabolic and B-cell receptor signaling pathways were activated in trastuzumab-resistant cell lines. Trastuzumab-resistant cells also grew faster and showed morphological signs of EMT (epithelial–mesenchymal transition) [50,51]. In particular, the mTOR pathway showed an upregulation of key signaling components such as mTOR, AKT and RPS6KB1, whereas expression levels of AKT1S1, a mTOR inhibitor, were decreased. Importantly, trastuzumab-resistant cells lines were susceptible to the mTOR inhibitor AZD8055, which reduced the migration of the cells. Janjigian et al. confirmed that multiple mutations in significant pathways such as the RAS and PI3K pathways mark important resistance mechanisms to trastuzumab [43].

The PI3K pathway plays a crucial role in tumorigenesis in various entities. In gastric cancer, HER2 overexpression significantly correlates with pAKT expression [52,53]. In multiple analysis, the activation of the PI3K pathway was one key factor contributing to trastuzumab resistance. The detection of PIK3CA mutations in liquid biopsies was associated with primary resistance to trastuzumab, even though other studies found no association between trastuzumab resistance and PIK3CA mutations [41,54]. One mechanism resulting in overactivation of the PI3K pathway is the inactivation of PTEN. PTEN acts as a tumor suppressor gene by dephosphorylating PIP3, the product of PI3K and a messenger in various pathways inducing cell growth and migration. The loss of PTEN leads to the constitutive activation of the PI3K pathway and inhibits apoptosis and G_1_ cell cycle arrest [5,55]. PTEN loss is associated with primary resistance to trastuzumab and represents a negative prognostic factor [56]. Other mechanisms of PI3K activation and mediating trastuzumab resistance include the upregulation of NES1/KLK10, the proto-oncogene XB130 or the c-MET-associated MACC1 [57,58,59]. As a response to trastuzumab, the HER3 ligand heregulin (or neuregulin1β) is released by the metalloprotease ADAM 10 and mediates resistance to HER2-targeting agents. Benefitting from residual HER2 activity, heregulin induces resistance through HER3 and AKT activation and upregulates the PI3K pathway [44,60,61].

A better understanding of the mechanisms of resistance might offer new therapeutic options to overcome such resistance, like the additional use of mTOR inhibitors or the combination with trastuzumab to delay resistance [51]. However, in the clinical setting, mTOR inhibition with everolimus could not significantly improve overall survival compared to BSC in pretreated advanced gastric cancer [62]. This might be due to compensatory PI3K upregulation by the phosphorylation of Akt that diminishes the anti-tumor effect of monotherapy with mTOR inhibition [63]. The PI3K pathway might therefore be another promising target [53]. In vitro, the PI3K inhibitor copanlisib and the MEK inhibitor refametinib could reinstall sensitivity to trastuzumab and lapatinib in breast cancer cells with acquired resistance to either trastuzumab and/or lapatinib [64,65]. Accordingly, for HER2-positive gastric cancer cell lines, the combination of copanlisib and trastuzumab demonstrated more anti-proliferative effects than either of the inhibitors alone. Importantly, the combination of trastuzumab and refametinib or copanlisib showed synergistic effects in some cell lines, and trastuzumab plus copanlisib even overcame resistance to either drug as a single agent in one cell line [53].

For clinical practice, it is crucial to monitor the success of HER2-targeting drugs. Liquid biopsies can be one way to identify resistance to therapy. HER2 somatic copy number alterations (SCNAs) in plasma predicted tumor shrinkage and progression more reliably than carcinoembryonic antigen levels. Moreover, patients with primary resistance to trastuzumab showed high HER2 SCNAs upon progression, whereas HER2 SCNAs decreased in patients with acquired resistance compared to baseline. Trastuzumab resistance was often accompanied by NF1 and ERBB2/4 gene mutations detectable in liquid biopsies [54].

Metabolic reprogramming plays a critical role in tumor cells, enabling cell growth and proliferation. Gastric cancer cells resistant to trastuzumab exhibit increased glycolysis, and 6-phosphofructo-2-kinase (PFKFB3) was shown to activate glycolytic pathways, resulting in distinct changes in the tumor microenvironment [57,66].

### HER2-Positive or Not?

Besides resistance mechanisms, HER2 heterogeneity represents an obstacle when targeting HER2. The definition of HER2 positivity relies on a pathological evaluation of tumor tissue samples. HER2-positive samples are commonly defined as IHC score 2+ and ISH-positive or IHC score 3+. For FISH assessment, a HER2:CEP17 (centromeric probe 17) ratio of ≥2 is defined as positive for HER2 amplification [67]. Due to the higher occurrence of glands in the gastric epithelium, incomplete (U-shaped or basolateral only) membranous staining is more common in gastric cancer than in breast cancer. Pathological HER2 scoring systems differ accordingly between the two entities, as for gastric cancer, samples with only basolateral staining in ≥10% of the cells are also considered positive. Because of the heterogeneity of HER2 expression in gastric cancer, the 10% cut-off was replaced by the absolute number of five cohesive cells with strong reactivity in biopsy specimens [67,68].

In a comparative analysis of 1414 cases of whole-tissue sections and 595 cases of tissue microarrays, samples scored 3+ stained ≥ 50% of the tumor area in over 90% of cases, whereas only 40% of 2+-scored samples achieved the same. Staining in less than 50% of the tumor area worsened the prognosis [12]. Even though the definition of heterogeneity in HER2 scoring is not clear, heterogeneity seems to be foremost relevant in lower IHC stained tumors [2]. In the HER2 screening data of the ToGA trial, IHC2+ and 3+ stained tumors showed a benefit from trastuzumab regardless of staining variability. However, a numerical trend towards better results was observed in tumors with >30% cells stained than tumors with ≤30% staining [2]. Another small study observed a significant difference in response to trastuzumab due to heterogeneity: tumors with a homogenous expression of HER2 had better PFS and OS than heterogeneously HER2-expressing tumors [69]. The recent VARIANZ study observed major discrepancies in the HER2 testing of tumors between local and central assessment (deviation rate of 22.7%, mainly due to false-positive testing locally). These tumors showed mostly intermediate HER2 expression. Only patients with a centrally confirmed HER2 positivity (central HER2+ and local HER2+) benefitted from a trastuzumab-based treatment compared to only locally positive tested tumors. HER2 expression and the HER2:CEP17 ratio were higher and more homogenous among centrally and locally positive tested tumors. The authors argue, that HER2 thresholds might have to be reconsidered to reliably identify patients who benefit from trastuzumab, and calculated a minimum of 40% HER2+ cells and an amplification ratio of 3.0 as optimized criteria [70]. Recently, Janijigan et al. demonstrated that the response to trastuzumab is increased in patients with high levels of ERBB2 amplifications in sequencing [43].

Moreover, the method of specimen taking might influence HER2 testing. The screening data of the ToGA trial revealed a numerically higher rate of HER2-positive tested tumors in biopsies than surgical specimens (23.2% vs. 19.7%) [2]. Van Cutsem et al. attributed this effect to few biopsy samples taken and the different criteria for HER2 positivity in biopsies (five cohesive cells) vs. tissue samples (≥10% of the cells). The GASTHER1 study confirmed the importance of multiple biopsies to assess the correct HER2 status. Patients who were initially considered as HER2-negative showed a conversion rate of 8.7% in immediate re-biopsies [71]. In particular, IHC 1+- or 2+-positive tumors compared to IHC 0 were more likely to be HER2-positive in re-biopsies. In a larger study, 17.6% had a rescued HER2 positivity after reassessment. Of note, rescued HER2-positive tumors showed a prognostic relevance with worse PFS and OS in the subgroup of IHC 3+ tumors than initially positive tested tumors [72]. Hence, current guidelines recommend taking multiple biopsies to correctly assess suspicious lesions: the German S3 guidelines suggest at least eight biopsies, and the NCCN guidelines and ESMO guidelines recommend 6–8 and 5–8, respectively [20,73,74].

Nevertheless, sampling errors due to HER2 heterogeneity still impose an obstacle even though multiple biopsies are taken. One study compared tissue microarrays (TMAs) serving as a “biopsy procedure” to whole-tissue sections from the same paraffin blocks. Whilst the VARIANZ study observed discrepancies in central and local HER2 testing, mostly due to false-positive testing locally with a significant impact on survival upon trastuzumab-based therapy, this study found a higher false-negative rate of 24% and only a minor false-positive rate of 3% [75]. The authors argue that unresectable gastric cancers might not receive HER2-directed therapy due to sampling errors in performed biopsies. Both studies considered the heterogeneity of HER2 expression as cause for discrepancies in HER2 testing.

Another aspect of spatial heterogeneity in HER2-positive gastric cancer is not only the discrepancies within the primary tumor but also varying HER2 expressions between different tumor lesions. In a small number of patients in the GASTHER1 trial, recurrent or metastatic (especially liver metastases) tumor lesions were HER2-positive in 5.7% even though the primary tumor was tested as HER2-negative [71]. Multiregion sequencing of the primary tumor and metastatic sites revealed differences in the genomic profiling, including oncogene amplifications such as EGFR, ERBB2, MET and PIK3CA [76]. Autopsies upon failure of HER2-targeting therapy confirmed a heterogeneous distribution of driver amplifications in different tumor lesions [77]. The authors argued that the limited efficacy of HER2-directed therapy was also due to heterogeneity and rapid selection of sensitizing or resistance-related genomic alterations [77].

Altogether, biopsies seem to be insufficient to capture the heterogeneity and complexity of HER2-positive gastric cancer. cfDNA obtained from peripheral blood plasma could be a suitable option to complete tumor assessment and to determine the indicated targeted therapies. In a pilot study, cfDNA enabled the detection of genomic alterations that were not exhibited in the sequencing of the primary tumor [76]. Simultaneously, high concordance rates for targetable alterations between metastases and cfDNA were shown, rendering cfDNA as a possible diagnostic tool to overcome spatial heterogeneity between different tumor lesions without the need for excessive and invasive biopsies [76]. However, the sensitivity of ctDNA might be dependent on disease site and burden, as detection was shown to be lower in peritoneal metastases than in liver or lung lesions and in tumors with fewer lesion sites [78]. In one trial, only 60% of clinically HER2-positive tumors were found to be positive in ctDNA evaluation. The combination of ctDNA plus tissue NGS might be the most accurate way to avoid false-negative results. HER2 amplification by ctDNA-NGA and/or tissue NGS was also prognostically relevant upon HER2 targeting (OS 26.3 vs. 7.4 months) [78].

An interesting non-invasive approach to circumvent the obstacle of heterogeneity could be the use of ^89^Zr-trastuzumab PETs. The HER2-directed functional imaging could potentially predict therapy response prior to disease progression in conventional CT/MRI imaging [77]. A pilot study showed that ^89^Zr-trastuzumab-PET was able to detect tumor lesions, especially in the bones [79]. In contrast, ^18^F-FDG-PET might be more sensitive for tumor manifestations in the lymph nodes. The HER2-PET is also limited in evaluating hepatic lesions due to high background signals. Furthermore, HER2-PET reflects HER2 heterogeneity throughout the entire body by varying tracer uptake between different tumor lesions, and might help to guide HER2-directed therapy and clinical decision making. The biomarker analysis of a phase II study including ^89^Zr-trastuzumab PETs and serial ctDNA demonstrated that the combination of these non-invasive assessments may predict PFS [80]. Concerning ctDNA, the clearance of all tumor-matched alterations was associated with a benefit in PFS.

## 4. New Perspectives in HER2 Targeting

HER2 heterogeneity and resistance mechanisms to HER2-targeting agents limit the efficacy of standard-of-care treatment with trastuzumab and T-DXd. Further attempts at HER2 targeting have been made to circumvent these obstacles.

### 4.1. Monoclonal and Bispecific Antibodies

**Pertuzumab** is a monoclonal antibody targeting extracellular domain II of HER2, preventing the formation of heterodimers such as the highly mitogenic HER2/HER3 dimer (see Table 2) [7]. In the JACOB trial, adding pertuzumab to first-line chemotherapy plus trastuzumab in metastatic HER2-positive esophagogastric cancer failed to significantly improve overall survival in spite of a positive trend in PFS and ORR. Moreover, the therapy showed more grade ≥ 3 diarrhea and nausea and led to a greater dose reduction of the chemotherapy [81].

**Margetuximab** is an Fc-modified chimeric antibody that increases the affinity to activated FcR (CD16A; FcγRIIIa) and reduces its affinity for inhibitory FcR (CD32B; FcγIIb). It enhances ADCC and boosts T-cell activity and PD-L1 expression [28]. In 2020, margetuximab was granted orphan drug status for esophagogastric cancer by the FDA. The MAHOGANY trial investigated margetuximab plus retifanlimab, a monoclonal antibody against PD-1, for first-line therapy of HER2/PD-L1-positive esophagogastric cancer. Compared to historical controls, margetuximab/retifanlimab showed a favorable toxicity in Cohort A. Given that this was a chemotherapy-free option, objective response rates were also improved. Nevertheless, the enrolment of the study was discontinued due to business reasons of the sponsor [82].

New developments for monoclonal antibodies also include, e.g., the combination of trastuzumab and **BI-1607**. BI-1607 is a monoclonal antibody against CD32b or FcγIIb and aims to enhance trastuzumab efficacy and escape resistance against it (NCT05555251). A phase I/IIa trial (CONTRAST) is currently ongoing for advanced solid HER2-positive tumors after failure of trastuzumab and HER2-targeting ADCs.

Furthermore, **HLX22** is tested in combination with trastuzumab and chemotherapy (XELOX) for first-line therapy of advanced HER2-positive gastric cancer (NCT04908813). HLX22 is a novel monoclonal antibody against domain IV of HER2. The targeted epitope is different compared to trastuzumab, enabling the simultaneous binding of both antibodies [83]. In a phase I study, HLX22 was tolerable and showed moderate anti-tumor activity after the failure of standard therapy lines in HER2-positive advanced solid cancer [83].

**Bispecific antibodies** such as, e.g., KN026 and zanidatamab (ZW25) are subjects of ongoing clinical studies and bind extracellular domains II (pertuzumab-binding site) and IV (trastuzumab-binding site) [28]. **KN026** exhibited promising anti-tumor activity in a phase II trial in HER2-expressing (ORR 56%) and HER2-low tumors (ORR 14%) after failure of one line of standard treatment [84].

**Zanidatamab** induces ADCC, HER2 internalization and downregulation, the inhibition of cell signaling, and tumor growth and shows superior anti-tumor activity in preclinical models compared to trastuzumab plus pertuzumab [85]. In untreated advanced or metastatic gastroesophageal adenocarcinoma, zanidatamab plus chemotherapy achieved an ORR of 68.2% and a DCR of 90.9% in a phase II study [86]. While guaranteeing a manageable safety profile, the most common treatment-related adverse events were diarrhea (grade 3 in 43%), vomiting and hypokalemia. The ongoing phase III HERIZON-GEA-01 trial evaluates the combination of zanidatamab and chemotherapy ± the checkpoint inhibitor tislelizumab for the first-line treatment of HER2-positive esophagogastric cancer [87].

**Cinrebafusp alfa**, a bispecific antibody against HER2 and the co-stimulatory immune receptor 4-1BB on T cells, has shown tolerability and anti-tumor activity in a phase I trial and is currently under investigation for second-line therapy of HER2-positive (high and low) gastric cancer [88].

### 4.2. Antibody–Drug Conjugates (ADCs)

To date, T-DXd remains the only authorized ADC for HER2-positive gastric cancer. Contrary to the results in breast cancer, **trastuzumab-emtansin** (T-DM1) was unsuccessful in esophagogastric cancer. In the phase II/III GATSBY trial, T-DM1 was not superior to conventional chemotherapy with taxane in previously treated HER2-positive advanced gastric cancer (OS: 7.9 vs. 8.6 months) (s. Table 1) [89]. The addition of capecitabine to T-DM1 in a phase I/II study also did not improve the ORR significantly [90]. T-DM1 is an ADC linking trastuzumab to the potent microtubule inhibitor emtansine (DM1), a derivative of maytansine [7,89]. Reasons for the failure of T-DM1 could be its lower drug-to-antibody ratio (3.4:1), the less potent payload and its poorer membrane permeability [29,30,33]. In contrast to T-DXd, the payload of T-DM1 remains attached to its linker even after degradation in the tumor cell. T-DM1 is therefore less capable of killing bystander tumor cells than T-DXd, which contributes significantly to the success of T-DXd in even heterogeneously HER2-expressing tumors. Moreover, reassessment of HER2 status was not required before enrolment in the GATSBY trial, so it has to be assumed that some of the tumors lost their HER2 expression after receiving prior HER2-directed therapy.

Other ADCs are currently under investigation. For **disitamab-vedotin** (RC48), an anti-HER2 monoclonal antibody–MMAE conjugate, a phase II trial with 179 patients showed an ORR of 24.8%, as well as a PFS and OS of 4.1 months and 7.9 months, respectively [91]. A previous phase I study including various solid tumors, but mostly (82.5%) gastric cancer, indicated similar responses in HER2-high and HER2-low expressing tumors [92]. Another phase I trial is currently recruiting patients to evaluate disitamab-vedotin plus the PD-1 inhibitor toripalimab for first-line treatment of HER2-low expressing advanced gastric cancer (NCT06078982). Furthermore, a phase II/III trial evaluates disitamab-vedotin plus chemotherapy, toripalimab and trastuzumab in HER2-expressing (high vs. low) tumors (NCT05980481). An overview of other ADCs currently under investigation in clinical trials is given in Table 3.

### 4.3. Small-Molecule Inhibitors

As an oral dual HER2/EGFR tyrosine kinase inhibitor, **lapatinib** could provide anti-cancer efficacy in pretreated and trastuzumab-resistant breast cancer patients, and the combination of trastuzumab and lapatinib improved the outcome in the neoadjuvant and metastatic setting [93,94,95]. In vitro, anti-tumor activity was shown in gastric cancer [96]. Lapatinib was able to inhibit cross-signaling with IGF1 in solid cancer and breast cancer, providing a rationale for its use upon progression to trastuzumab [97,98].

However, despite these promising preclinical data, lapatinib has only achieved disappointing results in the clinical setting for gastric cancer. In the TRIO-013/LOGiC trial, the combination of lapatinib and CAPOX for first-line therapy of advanced gastric cancer could not significantly improve OS and PFS, even though the response rates were higher than in the control arm. Differences in the effect of lapatinib in region and age were observed, with a prolongation of OS in Asian and younger patients [99]. Insufficient activity and increased toxic sides effects (mainly diarrhea) were confirmed in other studies evaluating lapatinib for second-line treatment [100,101]. In the TyTAN study, the addition of lapatinib to paclitaxel vs. paclitaxel alone in second-line treatment of advanced gastric cancer led to better results regarding ORR, but did not translate into a significant improvement in OS or PFS (see Table 1) [100]. Again, better results were observed in Asian patients and, contrary to the results of the LOGiC trial, were dependent on HER2 IHC status, with a greater effect on HER2 3+ tumors.

Liquid biopsies with cfDNA detecting the ERBB2 copy number were able to predict the response and sensitivity to lapatinib in a small phase II study combining lapatinib with CAPOX in the neoadjuvant setting [102]. Resistance to lapatinib is linked to the rephosphorylation and reactivation of EGFR and HER3 kinases, which can be delayed by combining anti-HER2 therapies in vitro [103]. Combining HER2-targeting therapies like lapatinib and trastuzumab might therefore be an option that is already proven to be effective in breast cancer. Accordingly, in gastric cancer cell lines and xenografts, a synergistic effect of trastuzumab and lapatinib was demonstrated [104].

Altogether, monotherapy with lapatinib has not provided convincing clinical results so far. Other small molecules inhibiting transphosphorylation or their combination might lead to more success and are currently under investigation [28].

The irreversible pan-HER (HER1, 2 and 4) inhibitor **afatinib** induced a sustained inhibition of EGFR and HER3 as sufficiently as the dual blockade of lapatinib and trastuzumab and provided a durable tumor regression in vitro [77,103,104]. In a small study with pretreated metastatic GC (including at least one line with trastuzumab), afatinib monotherapy showed modest clinical benefit with a PFS of 2 months and an OS of 7 months [77]. Whilst the co-amplification of ERBB2 and EGFR demonstrated a clinical benefit, tumors with KRAS, PIK3CA or NF1 mutations progressed rapidly. MET amplification also led to afatinib resistance. A modest anti-tumor effect was further described upon dual inhibition with trastuzumab and afatinib, even for ERBB2-amplified tumors lacking an EGFR amplification [77]. In the AGAPP trial, on the other hand, afatinib plus chemotherapy for the first-line treatment of metastatic GC showed no further advantage compared to present regimens, with a median PFS of 5.0 and an OS of 8.7 months. However, it must be taken into account that only 12.7% of the tumors were HER2-positive [105].

With **tucatinib**, a new orally available, highly selective and reversible HER2-targeting small-molecule tyrosine kinase inhibitor is on the horizon with promising anti-tumor activity in gastric cancer xenograft models and CRC cancer if combined with trastuzumab [106,107]. The ongoing MOUNTAINEER02 (NCT04499924) trial is investigating tucatinib plus trastuzumab, ramucirumab and paclitaxel in second-line treatment of advanced HER2-positive gastric cancer after progression to anti-HER2 targeted first-line therapy [108]. A phase I/II trial evaluating tucatinib and trastuzumab as first-line therapy is also in progress (NCT04430738). Combinations with checkpoint inhibition (pembrolizumab) and chemotherapy (FOLFOX/CAPOX) are available in this trial.

Dual HER2 inhibition is also under investigation using the pan-HER inhibitor **neratinib** and trastuzumab plus chemotherapy in the first-line treatment of HER2-positive gastroesophageal cancer (NCT06109467).

## 5. Other Developments in HER2 Targeting 

Genetically engineered T cells with **chimeric antigen receptors (CARs)** have led to major success in hematologic neoplasms by coupling an extracellular antigen-identifying domain to an intracellular stimulating domain that activates the CAR T cells [109]. However, targeting solid tumors with CAR T cells is a challenge due to antigen heterogeneity, infiltration difficulties and the immunosuppressive tumor microenvironment in solid cancer [109]. So far, HER2-targeting CAR T cells exhibited persistent anti-tumor activity in vitro and in vivo [110], and various phase I studies evaluate HER2-directed CAR T cells for solid tumors, including gastric cancer (see Table 3).

Other approaches include the modification of T cells. **T-cell antigen couplers (TACs)** are a novel technology to modify T cells in order to recognize HER2+ tumors. The TAC receptor recognizes tumor cells and co-opts with the endogenous TCR, mediating a more efficient anti-tumor response with less toxicity than CAR T cells [111]. In the phase I/II TAC01-HER2 trial, TACs showed manageable safety and promising clinical activity in various HER2+ tumors, including gastric and gastroesophageal junction cancer [112]. In the ongoing trial, TACs are combined with pembrolizumab (see Table 3).

**Cancer vaccines** provide tumor neoantigens (whole cells or characteristic components) to prime T and B lymphocytes against tumor epitopes [113]. In case of HER2-overexpressing gastric cancer, the IMU-131/HER-Vaxx was tolerable and safe in a phase I study and provided humoral, cellular and clinical responses in a dose-dependent manner. The vaccine consists of three fused B-cell epitopes from the HER2 extracellular domain coupled to the diphtheria toxoid and combined with the adjuvant montanide [114]. In the phase II HERIZON study, CAPOX plus HER-Vaxx improved the outcome compared to CAPOX alone in HER2-positive gastric adenocarcinoma naïve to HER2-targeting therapy [115]. The addition of HER-Vaxx resulted in a 42% survival benefit (HR 0.58, *p* = 0.066) and a median OS of 13.9 months (vs. 8.3 months for chemotherapy alone). HER-Vaxx is currently being evaluated in a phase II trial upon progression to trastuzumab (NCT05311176).

Another B-cell epitope HER2 vaccine is being tested in China (NCT05315830), and recently, a B-cell- and monocyte-based immunotherapeutic vaccine (BVAC-B) transfected with recombinant HER2 showed activation of immune cells and manageable toxicity. However, the clinical tumor response was limited in heavily pretreated patients [116].

**Immune modulation** is also achieved by CYNK-101, a human placental hematopoietic stem/progenitor cell-derived NK cell product. By expressing a variant of CD16, FcyRIII, it enhances ADCC. Dose-limiting toxicities are evaluated in a phase I trial, treating patients with CYNK-101, pembrolizumab and trastuzumab after lymphodepletion (NCT05207722).

Furthermore, the ASPEN-01 trial investigated the fusion protein evorpacept or ALX148 in various solid tumors. ALX148 blocks CD47, which is a surface protein acting as an immune checkpoint by suppressing tumor cell phagocytosis and innate immune functions. In preclinical studies, ALX148 enhanced phagocytosis as well as trastuzumab-associated tumor inhibition and elicited innate and adaptive anti-tumor responses [117]. In the ASPEN-01 study, 19 patients with HER2-positive gastric and gastroesophageal junction cancer received ALX148 and trastuzumab upon progression to HER2-direceted therapy. An ORR of 21%, a PFS of 2.2 months and an OS of 11.1 months were reported. ALX148 received fast-track designation by the FDA for development in HER2-postive gastric or gastroesophageal junction cancer [118]. The ongoing ASPEN-06 study evaluates the use of ALX148 plus trastuzumab and standard second-line therapy (ramucirumab/paclitaxel) in further treatment lines (NCT05002127).

A completely different approach to HER2-targeting is pursued with the use of CAM-H2 (131I-SGMIB anti-HER2-VHH1), a radioligand directed at HER2. A phase I study showed its tolerability and uptake into HER2-overexpressing lesions [119]. A dose escalation phase I/II study upon progression to anti-HER2 treatment is underway (NCT04467515).

Altogether, these new agents bare the potential to overcome resistance to trastuzumab in HER2-positve gastric cancer. T-DXd is, to date, the only available HER2-targeting therapeutic option upon progression to trastuzumab. Even though HER2 downregulation took place during trastuzumab therapy, T-DXd might be effective due to its bystander killing effect. Other ADCs such as disitamab-vedotin might lead to comparable results. Combining different therapeutical strategies such as different monoclonal antibodies, TKIs or immune modulators could also be an option to re-enable trastuzumab efficacy. Even though lapatinib has not led to success so far, other TKIs (afatinib and tucatinib in combination with trastuzumab) have shown promising results even upon progression to trastuzumab in vitro and in vivo [77,108,120]. A multitude of ongoing clinical studies (see Table 3) include patients in second and further lines. These studies will demonstrate whether new therapeutic agents are still effective after standard first-line therapy with trastuzumab and can potentially overcome resistance to trastuzumab. 

## 6. Perioperative Treatment of HER2 Gastric Cancer

After the introduction of trastuzumab and T-DXd into the therapeutical landscape of metastatic and unresectable HER2-positive gastric cancer, efforts are made to implement HER2 targeting into the perioperative treatment of locally advanced gastric cancer to further improve surgical und survival outcomes. Standard therapy for ≥T2 and/N+ tumors consists of perioperative chemotherapy with FLOT (docetaxel, 5-fluorouracil, leucovorin, oxaliplatin). In the FLOT-4 study, FLOT generated pCR rates of 16%, a median DFS of 30 months and a median OS of 50 months [121,122].

The phase II single-arm HERFLOT study demonstrated that the addition of trastuzumab to perioperative chemotherapy with FLOT results in a pCR rate of 21.4% [123]. This is of great interest, as high pCR rates are associated with favorable outcome [124,125,126]. In the HERFLOT study, only intestinal subtypes according to Laurén benefitted from trastuzumab compared to diffuse subtypes (pCR rate 33.3% vs. 0%). The median DFS was 42.5 months.

Similar to the JACOB trial in a metastatic setting, the following phase II PETRARCA study combined a dual HER2 blockade of trastuzumab and pertuzumab with perioperative FLOT [127]. In 81 patients, FLOT/pertuzumab/trastuzumab significantly improved the pCR rate (35% vs. 12%, *p* = 0.02) and survival rates at 24 months (DFS 70% vs. 54%; OS 84% vs. 77%) compared to FLOT alone. However, the addition of pertuzumab and trastuzumab led to an unfavorable toxicity profile. Higher rates of diarrhea as well as leukopenia and more dose modifications were described. The study was terminated prematurely.

The three-arm phase II EORTC INNOVATION study confirmed the great toxicity of dual HER2 blockade in the perioperative setting [128]. Perioperative chemotherapy (cisplatin/capecitabin in 42.2% and FLOT in 46.6%) was compared to trastuzumab or trastuzumab plus pertuzumab. A significant advantage in pathological response was only observed for the addition of trastuzumab (major pathologic response corresponding to <10% residual tumor in 37.0% vs. 23.3%, *p* = 0.099) and not for dual inhibition with trastuzumab and pertuzumab (26.4% vs. 23.3%, *p* = 0.378). This could be due to the toxicity of the dual inhibition, which led to less completion of the neoadjuvant treatment. Although the primary endpoint analysis did not meet the pre-specified criteria of efficacy for the combination of chemotherapy, trastuzumab and pertuzumab (increase in major pathologic remissions from 25% to 45%), the combination of chemotherapy and trastuzumab showed promising major pathologic response rates (mpRRs), especially with FLOT as the chemotherapy backbone (20% improvement in mpRR versus chemotherapy alone; 53% vs. 33%). EFS and OS data are still pending.

In Asian countries, gastrectomy and adjuvant chemotherapy alone represents the standard-of-care treatment. Preoperative chemotherapy is only applied in case of bulky lymph node disease [129,130]. For the latter, the combination of trastuzumab, S-1 and cisplatin preoperatively and S-1 adjuvant was feasible and safe in the TRIGGER study [131]. Radiological and pathological responses favored the trastuzumab group.

In summary, the combination of trastuzumab and perioperative chemotherapy showed promising results for increased pCR and survival rates, whereas dual inhibition might be too toxic. Nevertheless, more data are needed in this indication, and the use of anti-HER2 treatment in the perioperative setting is only recommended within clinical trials.

Recently, the synergism between ICI and HER2-targeted therapy further improved response and survival rates in the KEYNOTE-811 [24]. Multiple phase II trials are currently investigating the combination of ICI and trastuzumab plus chemotherapy for the perioperative setting (see Table 4). Other approaches in the perioperative setting include vaccination after standard adjuvant treatment with the pDNA-based AST-301 encoding HER2 (NCT05771584) or the use of ADCs (NCT05034887, NCT06155383).

## 7. Conclusions

In summary, HER2 remains an important biomarker for the treatment of gastric cancer. With T-DXd, significant progress has been made for the first time since the introduction of trastuzumab into the palliative treatment of metastatic gastric cancer. Other drugs are underway to further improve clinical outcomes and to face the challenges of-anti HER2 treatment, such as resistance mechanisms and HER2 heterogeneity.

Besides HER2, other important biomarkers make their way into clinical practice: The Claudin18.2-directed monoclonal antibody Zolbetuximab showed promising results in the SPOTLIGHT and GLOW studies [132,133]. The monoclonal antibody against FGFR2b, Bemarituzumab, is the subject of ongoing phase III studies (NCT05052801, NCT05111626). The treatment of gastric cancer will therefore involve further steps towards personalized therapy.

## Figures and Tables

**Figure 1 cancers-16-01336-f001:**
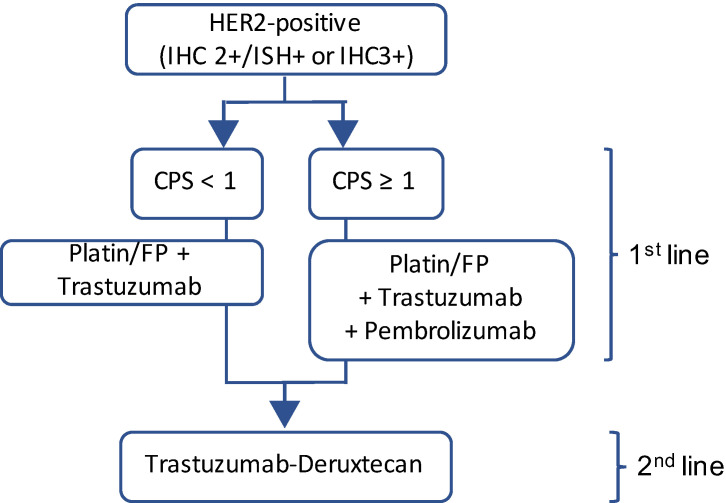
HER2 directed therapy in metastatic HER2-positive gastric cancer. FP = Fluoropyrimidine.

**Table 1 cancers-16-01336-t001:** Randomized phase II or III trials for second-line therapy of HER2-positive gastric cancer.

Study	*n*	Phase	Therapy	Primary Endpoint	Remark
DESTINY-Gastric01	187	II	T-DXd vs. chemotherapy	ORR 51% vs. 14% (*p* < 0.001)	SOC
T-ACT	91	II	trastuzumab + pacli vs. pacli	PFS 3.7 vs. 3.2 mo (ns)	unsuccessful
TyTan	261	III	lapatinib + pacli vs. pacli	OS 11 vs. 8.9 mo (ns)	unsuccessful (see below)
GATSBY	345	III	T-DM1 vs. taxane	OS 7.9 vs. 8.6 mo	unsuccessful (see below)

T-DXd = trastuzumab-deruxtecan; ORR = objective response rate; SOC = standard of care; mo = months; pacli = paclitaxel; cap = capecitabine; T-DM1 = trastuzumab-emtansine.

**Table 2 cancers-16-01336-t002:** Overview of selected HER2-targeting drugs.

Substance	Target	Remark
**Monoclonal antibodies**		
Trastuzumab Pertuzumab Margetuximab	extracellular domain IV of HER2 extracellular domain II of HER2 extracellular domain IV of HER2	
		HER2/HER3 dimerization modified FcR
**Bispecific antibodies**		
Zanidatamab KN026	Extracellular domain II and IV Extracellular domain II and IV	
**Antibody–drug conjugate (ADC)**		
Trastuzumab-Deruxtecan	HER2	exatecan-conjugated
Trastuzumab-Emtansin	HER2	emtansine-conjugated
Disitamab-vedotin	HER2	MMAE-conjugated
**Tyrosine kinase inhibitors (TKIs)**		
Lapatinib	HER2/EGFR	Small molecule
Tucatinib	Highly selective for HER2	Small molecule
Afatinib	Irreversible Pan-HER (HER1, 2 and 4)	Small molecule
Neratinib	Irreversible Pan-HER (HER1, 2 and 4)	Small molecule
**Others**		
Evorpacept/ ALX148	CD47	Fusion protein

**Table 3 cancers-16-01336-t003:** Selection of ongoing HER2-targeting studies in advanced gastric cancer.

Study	Phase	Drug	Therapy Line	Remark	Status	Primary Endpoint
**Monoclonal Antibodies**						
NCT05555251 CONTRAST	Ib/II	BI-1607	>2 L	Fc-Engineered mAB against CD32b + T	recruiting	safety, dose escalation
NCT04908813	II	HLX22	1 L	+ T + CAPOX vs. placebo	recruiting	PFS, ORR
**Bispecific Antibodies**						
NCT03929666	II	Zanidatamab	1 L	+ CTx	recruiting	safety, ORR
NCT05152147 HERIZON-GEA-01	III	Zanidatamab	1 L	+ CAPOX/ FP ± Tislelizumab vs. T + CAPOX/ FP	recruiting	PFS, OS
NCT05427383	II/III	KN026	≥2 L	+ Pacli/ Docetaxel/ Irinotecan vs. placebo	recruiting	PFS, OS
NCT05190445	II	Cinrebafusp alfa	≥2 L	± Ram + Pacli ± Tucatinib	active	ORR
**Tyrosine kinase inhibitors (TKIs)**						
NCT04499924 MOUNTAINEER02	II/III	Tucatinib	≥2 L	± T + Ram + Pacli vs. placebo + Ram + Pacli	active	OS
NCT04430738	Ib/II	Tucatinib	1 L	+ T ± FOLFOX/CAPOX, ± Pem	recruiting	dose escalation
NCT06109467	II	Neratinib	1 L	+ T + Pem + FOLFOX	recruiting	ORR
**Antibody–drug conjugate (ADC)**						
NCT04379596 Destiny-Gastric03	Ib/II	Trastuzumab Deruxtecan	1 L	± CTx ± ICI	recruiting	dose escalation (I), ORR (II)
NCT04704934 Destiny-Gastric04	III	Trastuzumab Deruxtecan	2 L	vs. Ram + Pacli	recruiting	OS
NCT06078982	I	Disitamab Vedotin (RC48)	≥2 L	+ Toripalimab HER2-low	recruiting	ORR
NCT05980481	II/III	Disitamab Vedotin (RC48)	1 L	+ Toripalimab ± T ± CAPOX	recruiting	safety
NCT05141747	II	MRG002	≥2 L	HER2-positive and low	recruiting	ORR
NCT04492488	I/II	MRG002	last line		recruiting	dose escalation, ORR
NCT05671822	Ib/II	SHR-A1811 (Trastuzumab Rezetecan)	≥2 L	± ICI ± X ± CAPOX	recruiting	dose escalation, ORR
NCT03821233	I	Zanidatamab Zovodotin (ZW49)	last line	HER2-positive advanced solid tumors	active	safety, tolerability
NCT03255070	I	ARX788	last line	HER2-positive advanced solid tumors	active	safety, ORR
**Immune modulation**						
NCT03740256 VISTA	I	HER2 CAR plus CAdVEC (oncolytic adenovirus)	last line	basket trial HER2-positive solid tumors	recruiting	safety
NCT04511871	I	HER2 CAR	last line	HER2-positive solid tumors	recruiting	dose escalation
NCT04660929	I	HER2 CAR macrophage	last line	HER2-positive solid tumors ± Pem	recruiting	safety
NCT04727151 TACTIC-2	I/II	TAC T cells (TAC01-HER2)	≥2 L	HER2-positive solid tumors ± Pem	recruiting	safety (I) ORR, PFS, OS (II)
NCT05315830	I	HER2 tumor vaccine	further lines	± P/ F/ X	recruiting	safety
NCT05311176 nextHERIZON	II	IMU-131 (HER-Vaxx)	≥2 L	± Ram + Pacli ± Pem	recruiting	safety, ORR
NCT05207722	I/II	CYNK-101	1 L	+ Pem + T + CTx	active	dose escalation
NCT05002127 ASPEN-06	II/III	Evorpacept (ALX148)	≥2 L	+ T + Ram + Pacli vs. Ram + Pacli ± T	recruiting	ORR (II) OS (III)
**Divers**						
NCT04467515	I/II	CAM-H2	≥2 L	HER2-directed radioligand	recruiting	ORR

CTx = chemotherapy, F = 5-fluorouracil, ICI= immune checkpoint inhibition, P = cisplatin, Pacli = paclitaxel, Pem = pembrolizumab, Ram = ramucirumab, T = trastuzumab, X = capecitabin.

**Table 4 cancers-16-01336-t004:** Overview of trials for locally advanced HER2-positive gastric cancer.

Study	Phase	Intervention	Endpoint
NCT05504720 PHERFLOT	II	trastuzumab plus pembrolizumab plus FLOT	DFS, pCR
NCT05715931	II	trastuzumab plus toripalimab plus FLOT	pCR
NCT04819971	II	trastuzumab plus tislelizumab plus S-1/oxaliplatin/docetaxel	pCR
NCT05771584 CONERSTONE3	II	AST-301 vaccine after standard adjuvant treatment (Taiwan) in HER2 overexpression/low	Safety, immunologic efficacy
NCT05034887	II	T-DXd neoadjuvant in HER2 overexpression/low	MPR
NCT06155383	II	Disitamab vedotin plus toripalimab ± XELOX vs. XELOX	pCR

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
