# Peer review of "HER2-Positive Gastric Cancer and Antibody Treatment: State of the Art and Future Developments"

_cancers, 2024, doi:10.3390/cancers16071336_

Round 1

Reviewer 1 Report

Comments and Suggestions for Authors

The manuscript provides a comprehensive overview of the current landscape of HER2-targeted therapies in the treatment of gastric cancer. It explores the challenges associated with HER2 heterogeneity and resistance mechanisms. Then it discusses new perspectives in HER2 targeting, including monoclonal and bispecific antibodies, antibody-drug conjugates (ADCs), and small molecule inhibitors. Specifically, it highlights several promising developments, including the exploration of novel monoclonal antibodies like margetuximab and HLX22, bispecific antibodies like KN026 and zanidatamab, and ADCs beyond T-DXd. It also delves into the potential of small molecule inhibitors like lapatinib, afatinib, and tucatinib, detailing their efficacy and limitations in clinical trials. Additionally, it extends the direction to emerging strategies beyond traditional drug modalities, including genetically engineered T cells with chimeric antigen receptors (CAR-T cells), T cell antigen couplers (TAC), cancer vaccines, and immune modulation with agents like CYNK-101 and ALX148. The manuscript further investigates the incorporation of HER2-targeted therapies into the perioperative treatment of gastric cancer, examining the potential benefits of combining trastuzumab with chemotherapy and the challenges associated with dual HER2 blockade. The synergy between immune checkpoint inhibitors (ICI) and HER2-targeted therapy is discussed, along with ongoing clinical trials exploring these combinations. In conclusion, the manuscript emphasizes the ongoing efforts to improve outcomes in gastric cancer treatment through advancements in HER2-targeted therapies. While the manuscript covers various aspects of HER2-targeted therapies for gastric cancer, the following topics could be considered for elaboration:

1. Expand on the discussion of resistance mechanisms to HER2-targeted therapies and potential strategies to overcome resistance.

Reviewer 2 Report

Comments and Suggestions for Authors

Overall, this review is comprehensive and well written. However, there are a few minor remarks, as noted below.

Major comments

1.     Although the term "antibody" is mentioned in the title, the main topics are TKIs or ADCs targeting HER2. This reviewer would suggest that the title be changed.

2.     Paragraphs are changed in a couple of sentences. One paragraph should contain more sentences, leading to reduction of too many paragraphs.

3.     “Resistance to HER2-Targeting Drugs” should be “Resistance to Trastuzumab-Targeting Drugs”. If the authors discuss “Resistance to HER2-Targeting Drugs”, the resistance mechanisms of T-Dxd should be mentioned more. Reports regarding such resistance mechanisms have already published as for breast cancer field.

Minor comments

1.     The ToGA trial enrolled much more patients and a total of 584 HER2-positive patients were included. This should be described (Page 2).

2.     The proportion of PD-L1-negative patients in the CheckMate-649 should be revised (Page4).

3.     The T-ACT study was conducted in Japan. “Asian” should be “Japanese” (Page 4).

4.     “Irinotecan” should be “SN-38” (Page 4).

5.     In the Figure 1, the description of second-line chemotherapy should be revised since T-Dxd was not established as the standard of second-line treatments.

6.     The reference of the KEYNOTE-811, SPOTLIGHT, and GLOW should be added (Page 17).

Comments on the Quality of English Language

 The manuscript would be improved by a thorough English proofreading. There appear to be several minor errors. For example, “S1” should be “S-1” or “german” should be “German” (Page 2).
